# Stem Cell Responsiveness to Imatinib in Chronic Myeloid Leukemia

**DOI:** 10.3390/ijms242316671

**Published:** 2023-11-23

**Authors:** Rachid Lahlil, Anne Aries, Maurice Scrofani, Céline Zanetti, Desline Hennequin, Bernard Drénou

**Affiliations:** 1Institut de Recherche en Hématologie et Transplantation (IRHT), Hôpital du Hasenrain, 87 Avenue d’Altkirch, 68100 Mulhouse, France; ariesa@ghrmsa.fr (A.A.); drenoub@ghrmsa.fr (B.D.); 2Laboratoire d’Hématologie, Groupe Hospitalier de la Région de Mulhouse Sud-Alsace, Hôpital E. Muller, 20 Avenue de Dr. Laennec, 68100 Mulhouse, France

**Keywords:** CML, BCR-ABL, stem cells, VSELs, LVSELs, imatinib, leukemic stem cells, drug resistance, miRNA

## Abstract

Chronic myeloid leukemia (CML) is a clonal myeloproliferative disease characterized by the presence of the BCR-ABL fusion gene, which results from the Philadelphia chromosome. Since the introduction of tyrosine kinase inhibitors (TKI) such as imatinib mesylate (IM), the clinical outcomes for patients with CML have improved significantly. However, IM resistance remains the major clinical challenge for many patients, underlining the need to develop new drugs for the treatment of CML. The basis of CML cell resistance to this drug is unclear, but the appearance of additional genetic alterations in leukemic stem cells (LSCs) is the most common cause of patient relapse. However, several groups have identified a rare subpopulation of CD34^+^ stem cells in adult patients that is present mainly in the bone marrow and is more immature and pluripotent; these cells are also known as very small embryonic-like stem cells (VSELs). The uncontrolled proliferation and a compromised differentiation possibly initiate their transformation to leukemic VSELs (LVSELs). Their nature and possible involvement in carcinogenesis suggest that they cannot be completely eradicated with IM treatment. In this study, we demonstrated that cells from CML patients with the VSELs phenotype (LVSELs) similarly harbor the fusion protein BCR-ABL and are less sensitive to apoptosis than leukemic HSCs after IM treatment. Thus, IM induces apoptosis and reduces the proliferation and mRNA expression of Ki67 more efficiently in LHSCs than in leukemic LVSELs. Finally, we found that the expression levels of some miRNAs are affected in LVSELs. In addition to the tumor suppressor miR-451, both miR-126 and miR-21, known to be responsible for LSC leukemia-initiating capacity, quiescence, and growth, appear to be involved in IM insensitivity of LVSELs CML cell population. Targeting IM-resistant CML leukemic stem cells by acting via the miRNA pathways may represent a promising therapeutic option.

## 1. Introduction

The BCR-ABL fusion protein kinase is a unique oncogenic driver of chronic myeloid leukemia (CML) by affecting hematopoietic stem cells. This fusion protein transforms healthy hematopoietic stem cells (HSCs) into leukemic hematopoietic stem cells (LHSCs) [1]. The use of the potent tyrosine kinase inhibitor imatinib (IM) as the first-line treatment of CML has significantly decreased patient mortality. However, several clinical challenges remain as some patients eventually experience resistance to IM. The mechanisms underlying IM resistance could include point mutations in the BCR-ABL tyrosine kinase domain or the increased expression of the BCR-ABL oncogene [2]. Nevertheless, BCR-ABL-independent mechanisms underlying the persistence of quiescent CML stem cells or the alterations of efflux/influx pumps have also been observed [3]. Since VSELs are permanently present closer to hematopoietic stem cells in the bone marrow and peripheral blood mostly in the quiescent state, as well as being able to differentiate toward different hematopoietic cell types in healthy individuals [4,5], we hypothesized that they might be impacted by the genetic abnormalities observed in CML and be responsible for the relapse and/or resistance to IM among patients under treatment. Indeed, it has been reported that primitive CML cells remain insensitive to IM despite the inhibition of BCR-ABL activity, and therapies that biochemically target this activity will not entirely eliminate CML stem cells, which may persist in patients despite prolonged treatment [6,7]. The reason for IM treatment failure in some patients is unclear; however, leukemic cells’ persistence could be due to a change in the expression levels of multiple genes implicated in cell resistance to drugs and specific microRNAs (miRNAs). Indeed, increasing evidence shows that miRNA expression is deregulated in hematological malignancies and is responsible for maintaining a leukemogenic state [8,9]. Specifically, miRNAs, circular RNAs (cirRNAs), and long non-coding RNAs have been shown to contribute to the development and progression of TKI resistance to treatment in CML cells through a diverse set of mechanisms and pathways [10]. Several groups have shown that a number of miRNAs affect the sensitivity of CML stem cells to TKI in vitro and, as a consequence, confer these cells resistance [11]. Except for the established role of some miRNAs in the quiescent state of VSELs [12], much remains to be understood regarding their implications in other processes.

We therefore decided to determine whether this leukemia is initiated only in HSCs or further upstream at the level of LVSELs by quantifying BCR-ABL mRNA expression levels. We also characterized their resistance in vitro to IM by measuring their sensitivity after treatment. We observed that this primitive population of LVSELs, i.e., Lin^−^CD34^+^CD45^−^CD133^+^, expresses the oncogenic fusion protein and, in the presence of IM, is less sensitive to apoptosis than LHSCs, Lin^−^CD34^+^CD45^+^. In addition, we hypothesized that mi-RNAs might mediate LVSELs’ insensitivity to IM. Consequently, by analyzing the expression levels of a set of miRNAs in LVSELs and comparing them to HSCs treated or not treated with IM, we found that miR-126, miR-21, and miR-451 expressions were dysregulated, indicating their implication in cell resistance to IM.

## 2. Results

### 2.1. BCR-ABL Expression in Leukemic LHSCs and LVSELs

To determine BCR-ABL expression in the two different subsets, Lin^−^/CD34^+^ stem cells were isolated using the magnetic beads method from the peripheral blood and/or the bone marrow (BM) biopsies of newly diagnosed patients who were not yet treated with IM or those who were in “remission” and follow-up after receiving TKIs for at least a few months and were under follow-up. For this study, around 36 patients were enrolled, 16 were newly diagnosed and 20 were under follow-up (Table 1). Their clinical features are summarized in Table 1. All patients with reported data were diagnosed with chronic–phase CML and tested positive for BCR-ABL translocation. The average age in the newly diagnosed and the follow-up groups was closer to 57 years (range: 25–81 years). The white blood cell count was significantly higher in the newly diagnosed group (92 ± 76 × 10^9^/L) compared to patients who were in follow-up (6 ± 2 × 10^9^/L). Additionally, the average platelet count at the time of initial diagnosis was 586 ± 560 versus 204 ± 91 × 10^9^/L in the follow-up group. Only minor clinical differences were observed for other patient characteristics between the two groups (Table 1). No patient in the follow-up group had splenomegaly at the time of sample collection.

The LHSCs Lin^−^CD34^+^CD45^+^, and LVSELs, Lin^−^CD34^+^CD45^−^CD133^+^ were then isolated via FCM sorting (Figure 1A), and their BCR-ABL expression was quantified using real-time RT-PCR and corrected to ABL mRNA levels. As indicated, the fusion protein was expressed in all cells derived from newly CML-diagnosed patients (Pt1 to Pt5) in both LHSCs and LVSELs present in the peripheral blood (Figure 1B). Conversely, when patients were under follow-up and BCR-ABL, expression was no longer detected in LHSCs (Pt6 to Pt9), and the same pattern was found in LVSELs. Similar results were observed when cells were isolated from the BM (Figure 1C). Thus, patients Pt10 to Pt21 were all positive for BCR-ABL expression in both populations of cells; however, patients Pt23 to Pt32 tested negative for the fusion protein, suggesting that BCR-ABL translocation is initiated upstream of HSCs, at least at the VSEL level.

### 2.2. Quantification of Lin^−^CD34^+^ Stem Cell Populations’ Responsiveness to IM

By using the method developed in our laboratory and described in [13], we first isolated and expanded the purified Lin^−^CD34^+^ stem cells of newly diagnosed patients, then treated them with 0.1, 1, and 5 µM of IM for 48 h. We then determined the relative percentages of the two subsets of stem cells (Figure 2A) versus lineage-negative cells using FCM. As shown in Figure 2B, we observed that the percentages of Lin^−^CD34^+^CD45^−^CD133^+^ (LVSELs) increased significantly over the total lineage-negative cells in a dose-dependent manner when compared to untreated cells. In contrast, the percentages of Lin^−^CD34^+^CD45^+^ LHSCs decreased drastically after treatment in comparison to lineage-negative cells. Interestingly, while LVSELs represented only 2.98 and 2.82% of LHSCs in untreated and 0.1 µM IM-treated cells, respectively, they reached 15% and 19.97% of HSCs in 1 µM and 5 µM IM-treated cells, respectively (Figure 2C). These results indicate that leukemic LVSELs are enriched compared to LHSCs after IM treatment.

### 2.3. Quantification of Stem Cell Survival

To determine if the reason for the gradual effects of IM treatment on cell numbers was related to the survival of different stem cell subsets, we analyzed cell survival in cells of newly diagnosed patients treated with 5 µM IM in comparison to untreated cells by assessing apoptosis and mortality using annexin V and 7ADD labeling, respectively. At 0 μM IM (control), the initial apoptosis was slightly greater in LVSELs (26.1%) than in HSCs (8.6%) (Figure 3A). However, after 48 h of treatment with 5 μM IM (Figure 3B), we observed that the rate of apoptosis increased more substantially in LHSCs (72.3%) than in LVSELs (37.3%), indicating a clear resistance of LVSELs to the IM treatment. As shown in Figure 3C, which presents the means of four independent experiments, LVSELs are significantly less sensitive to the IM treatment, suggesting their possible implication in CML patients experiencing IM relapse.

### 2.4. Quantification of Gene Expression

Real-time RT-PCR analysis of representative proliferative genes’ mRNA expression levels showed that after the IM treatment of newly diagnosed patients’ cells, the expression level of Ki67 was inhibited in LHSCs but not in LVSELs (Figure 4). This suggests that IM affected HSC proliferation more. However, the mRNA expression levels of the Cip/Kip family of cyclin-dependent kinase inhibitors, p57 and p21 genes, which are known as important tumor suppressor genes, were significantly induced after the IM treatment in LHSCs; in contrast, their expression levels were not significantly affected in treated LVSELs, suggesting that they are involved in ensuring HSC apoptosis following IM treatment. Indeed, p57Kip2 is inactivated in various types of cancer [14,15], and its selective expression sensitizes cancer cells to apoptotic agents such as cisplatin etoposide and staurosporine [16].

### 2.5. miRNA Expression in Lin^−^CD34^+^CD45^+^ and Lin^−^CD34^+^CD45^−^ CML Stem Cells

Since miRNAs have been shown to specifically contribute to CML patients’ sensitivity to TKI by affecting the survival of primitive stem cells [10], and LVSELs and LHSCs were found to show a difference in their survival in the presence of IM in this study, we asked whether miRNA expression could be deregulated in these two subpopulations. Among the miRNAs known as key regulators that exert their effects on CML LSC maintenance and/or resistance to TKI through diverse mechanisms, we selected miR-21 [14], miR-29b, miR-30e [17], miR-126 [18], miR-130a [19], miR-378 [20], miR-146a [21], miR-199a [22], miR-320b [23], miR-451 [24], miR-486 [25], and miR-494 to evaluate their expression levels in the subpopulations of expanded stem cells isolated from biopsies of newly diagnosed patients that were exposed or not exposed to IM. We first sorted defined subsets of LHSCs (Lin^−^CD34^+^CD45^+^) and LVSELs (Lin^−^CD34^+^CD45^−^) from untreated and treated CD34^+^ SCs isolated from patients affected by CML and not receiving TKI. We then extracted RNAs and determined the expression level of each miRNA in these two subpopulations. Interestingly, in untreated cells, miR-126 and miR-21 expression levels indicated that they were the most enriched miRNAs found in LHSCs (Figure 5A) and LVSELs (Figure 5B). However, the expression levels of miR-29b, miR-199a, miR-378, miR-30e, miR-494, miR-130a, miR-146a, miR-320b, and miR-486 were detected at lower levels and were equivalent in these two subpopulations (Figure 5A,B). The IM treatment appears to not significantly affect the expression of these miRNAs in the two subpopulations of cells (Figure 5A,B).

However, when comparing between IM-untreated LHSCs and LVSELs, miR-451 expression was significantly increased by 3.5-fold in LVSELs compared to LHSCs (Figure 6). In contrast, miR-199a was significantly downregulated by 0.4-fold in LVSELs compared to LHSCs (Figure 6). Finally, all other miRNAs tested showed no statistically significant differences in their expression levels between untreated LHSCs and LVSELs (Figure 6).

When cells from CML patients were treated with IM, we observed a significant increase in miR-126a, miR-21, miR-130a, and miR-451 expressions by 2.9-, 1.6-, 1.4-, and 3.1-fold, respectively, in treated LVSELs versus treated LHSCs (Figure 6). In contrast, the miR-199a expression levels were 0.6-fold significantly lower in treated VSELs than in treated LHSCs (Figure 6). However, miR-29b and miR-146a appeared to not be affected significantly by the IM treatment when compared to LVSELs and LHSCs (Figure 6). Taken together, these results suggest that miR-126, miR-451, and miR-21 may play a role in the observed LVSELs’ insensitivity to IM.

## 3. Discussion

CML is a clonal myeloproliferative disease that occurs in hematopoietic stem cells as a result of the constitutive expression of BCR-ABL. Although it has been established that CML origin starts in LHSCs, different studies have implicated a precursor that gives rise to other lineages [26]. Indeed, evidence suggests that BCR-ABL gene rearrangement could occur at or even before the level of hemangioblast progenitor cells, as it has been shown that the BCR-ABL fusion gene could be detected in blood and endothelial cells [27,28]. This led us to suggest LVSELs as pluripotent cells that might contribute to malignant hematopoiesis and endeavored to follow BCR-ABL expression in blood-circulating stem cells from CML patients. Our result indicated that this fusion protein was expressed in both LHSCs and LVSELs of all newly diagnosed patients. However, when patients were under follow-up and showed Philadelphia chromosome-negative disease, BCR-ABL was no longer detected either in LHSCs or in LVSELs. As circulating LVSELs were different from those present in the BM due to the permanent exchange of the latter with the medullary microenvironment of stem cell niche, we also monitored BCR-ABL expression in stem cells isolated from the BM of CML patients. Interestingly, we found that these stem cells were also affected by the fusion protein, suggesting that this leukemia could be initiated in both LHSCs and LVSELs before reaching circulation.

Like pluripotent stem cells, LVSELs nature, physiology, and resistance to chemotherapy could be different when compared to LHSCs. To date, a consensus view has emerged that some patients undergoing TKI therapy and achieving molecular response are not cured of CML and show signs of residual disease burden due to the persistence of quiescent subpopulations of leukemic stem cells in the BM [29,30]. We therefore proposed to determine the sensitivity to IM of these two subpopulations of leukemic cells (LHSCs and LVSELs) that are present in the BM of CML patients. We found that LVSELs are in vitro less sensitive to the treatment; thus, IM reduces HSC amounts more efficiently by being able to more effectively induce their apoptosis. These results are consistent with previous studies showing that CML-quiescent leukemic stem cells have a higher capacity to engraft in immunocompromised mice than bulk CD34^+^ cells [31], have stem cell properties (self-renewal), are resistant to apoptosis [32,33], are prone to genomic instability, and have impaired DNA damage responses [34,35]. Expression of the proliferative marker Ki-67 was higher in untreated LHSCs than in LVSELs, indicating that they are more likely to cycle than the latter. The IM treatment reduced this proliferative marker’s expression only in LHSCs, attesting to their sensitivity to IM. However, the expression levels of p57 and p21, which are known to negatively regulate the cell cycle and act as putative tumor suppressors [36,37], were upregulated in LHSCs but not in LVSELs after the IM treatment, which is in agreement with the observations that their activity is reduced in various types of human cancers. IM appears to restore p57 and p21 expressions in LHSCs but not in LVSELs.

We then examined whether LVSELs from CML patients were affected by dysregulated expression of miRNAs, which might contribute to their lower sensitivity to IM treatment. We therefore measured miRNA expression levels in LVSELs and LHSCs isolated from the BM of CML patients. Among the group of miRNAs that we studied and found their expression levels to be most enriched in LHSCs and LVSELs, miR-126 has been reported to regulate the self-renewal of LSCs in CML [18] and miR-21 is known as a biomarker of this disease’s progression, with expression increasing from the chronic to the blast crisis phase [38] (Figure 5). Interestingly, miR-126 and miR-21 could be involved in LVSELs resistance to IM in CML patient cells as their expression is increased similarly in treated LVSELs when compared to treated LHSCs (Figure 6). Our results were consistent with the finding of Zhang and colleagues that miR-126 expression followed the hierarchy of hematopoietic differentiation of human CML cells, with primitive hematopoietic stem cells or progenitors expressing higher levels of miR-126 than mature cells [18]. Moreover, these authors showed that the inhibition of BCR-ABL by TKI treatment caused an increase in miR-126 levels, which contributed to LSC persistence by regulating these cells’ dormancy and engraftment potential. On the other hand, Alves and colleagues showed that miR-21 was upregulated in IM-resistant CML K562 cells [39]. Furthermore, concurrent treatment with antagomiR-21 and IM markedly increased the efficacy of IM and apoptosis of CD34^+^ CML cells without affecting normal CD34^+^ cells by blocking the PI3K/AKT signaling pathway [40]. Therefore, as miR-126 and miR-21 show a significant increase in their expression in LVSELs compared to LHSCs after IM treatment, their implication in LVSEL insensitivity remains plausible. However, loss- or gain-of-function experiments in cell line models remain to be carried out in order to confirm these findings. In contrast, the tumor suppressor miR-130a, which is known to induce apoptosis and suppress cell proliferation [41], was less enriched, and its expression was induced moderately in LVSELs after the IM treatment. On the other hand, miR-451 has been reported to act as a tumor suppressor in many malignancies, and its expression level is downregulated in CML patients at diagnosis and in patients with hematological relapse compared to healthy donors [42]. However, in the present study, miR-451 expression was enriched in treated and untreated LVSELs compared to LHSCs but remained unaffected by the IM treatment in LVSELs, suggesting that miR-451 is insensitive to the IM treatment in LVSELs and that its expression did not reach a level that ensures the reversion of their insensitivity and their resistance to apoptosis. Our results also showed that miR-199a expression was downregulated in treated LVSELs compared to treated LHSCs. A previous study reported that miR-199a expression was significantly associated with IM drug resistance in CML patients [43]. miR-199a overexpression suppressed cell proliferation and sensitized CML patients to IM via downregulation of mTOR signaling [44]. Conversely, in acute myeloid leukemia, its downregulation is beneficial as patients with a high expression level of miR-199a have significantly worse overall and event-free survival than AML patients with low expression [45]. In conclusion, the expression levels of miR-126, miR-21, miR-451, and miR-199a in LVSELs could play a role in CML response to IM treatment. However, although the expressions of the other miRNAs tested (miR-146a, miR-30e, miR-320b, miR-378, miR-486, and miR-494) were not significantly modified, K562 leukemic cells stably transfected with miR-378 showed dramatically enhanced proliferation and resistance to the drug [46]. The forced expression of miR-29b in K562 cells inhibits their growth and induces apoptosis through the regulation of BCR-ABL protein by acting as an oncomiR [47]. On the other hand, downregulation of miR-494 is able to decrease TKI-induced apoptosis [11]. Taken together, our results suggest that LVSELs have a different physiology than other LSCs but LVSELs share properties like quiescence, pluripotency, self-renewal, immortality, plasticity, enrichment in side-population, mobilization, and resistance to oncotherapy [48]. In addition, these data support the possible implication of LVSELs as CML LSCs through miRNA expression dysregulation in disease persistence despite treatment. Given that the total eradication of CML LSCs during TKI treatment is yet unresolved, our results may be clinically relevant to patients with CML who are treated with a TKI by facilitating the development of new drugs specifically targeting miRNA expression in leukemic VSELs.

## 4. Materials and Methods

### 4.1. Stem Cell Expansion

Lin^−^CD34^+^ cells obtained from the bone marrow biopsies or blood of LMC patients, who had not received prior TKI treatment or under follow-up, were procured from GHRMSA hospital, purified by means of EasySep™ (STEMCELL Technologies, Grenoble, France) according to the manufacturer’s instructions, and expanded for 5 to 12 days in StemSpan^TM^ ACF medium (STEMCELL Technologies) supplemented with growth factors, stem cell factor (SCF; 100 ng/mL, R&D Systems, Noyal Châtillon sur Seiche, France), Flt3 ligand (FLT3-L; 100 ng/mL, R&D Systems), and thrombopoietin (TPO; 20 ng/mL, PEPROTECH) and UM171 (35 µM, STEMCELLS Technologies), as described in [13]. Viable Lin^−^CD34^+^CD45^−^CD133^+^ and Lin^−^CD34^+^CD45^+^ cells were then sorted and quantified.

### 4.2. Flow Cytometry Analysis

To quantify stem cells, bone marrow cells were stained with a mixture of lineage (Lin)-associating monoclonal antibodies (MoAbs) conjugated with fluorescein isothiocyanate (FITC). At the same time, V500-conjugated CD45 (Beckman Coulter, Villepinte, France), CD34 PE clone 8G12, a combination of allophycocyanin (APC)-conjugated MoAbs, and CD133 clone AC133 (Miltenyi Biotec, Bergisch Gladbach, Germany) were added for 30 min on ice. The labeled cells were washed and discriminated via FCM on the basis of cell size, granularity, absence of Lin and CD45 markers, and presence of CD34 and CD133. Apoptotic cells were quantified by labeling cells with Annexin V-APC (BD Biosciences, France). The FCM analysis and sorting were performed using a BD ARIA III instrument (BD Biosciences Le Pont de Claix, France). Data acquisition and analysis were conducted using BD-FACSDiva software (BD Biosciences, Le Pont-de-Claix, France).

### 4.3. RNA Isolation and Quantitative RT-PCR

Total RNA was isolated and prepared using an RNeasy Plus Mini Kit (Qiagen, Courtaboeuf, France) and quantified (DeNovix, Wilmington, DE, USA). RNAs were then reverse transcribed (RT) into complementary DNAs (cDNAs) using IScript Supermix (Bio-Rad, Marnes-la-Coquette, France) or RT SuperScriptTM VILO TM Master Mix (Life Technologies SAS, Courtaboeuf, Villebon-sur-Yvette, France) according to the manufacturer’s instructions. The relative gene expression of the obtained cDNAs was assessed by means of real-time PCR using SsoAdvanced Universal SYBR Green Supermix (Bio-Rad) or Master Mix TaqMan™ Fast Advanced (Life Technologies SAS) using a CFX96 Real-Time PCR System (Bio-Rad) and the following indicated primers (Table 2):

### 4.4. MicroRNA Quantification

Total RNA from stem cells was isolated using an miRNeasy Tissue/Cells Advanced MicroKit (QiAGEN, Courtaboeuf, France) according to the manufacturer’s protocols. The RNAs isolated from cells were reverse transcribed to cDNAs using the miCURY LNA RT Kit (QIAGEN, Courtaboeuf, France). UniSp6 RNA spike-in controls were added during cDNA synthesis to ensure the quality of the experiment. Real-time qPCR amplifications were performed for each RT reaction. These reactions were performed according to the manufacturers’ instructions using a miRCURY LNA miRNA SYBR Green PCR Kit (QIAGEN, Courtaboeuf, France) with the Bio-Rad CFX96^TM^ Real-time PCR Detection System (BioRad Laboratories, Marnes-la-Coquette, France). All primer sets were custom designed by the supplier. The primers used were miR-21-5p (YP00204230_),_ miR-29b-3p (YP00204679), miR-30e-5p (YP00204714), miR-126-3p (YP00204227), miR-130a-3p (YP002046658), miR-146a-5p (YP00204688), miR-199a-3p (YP00204536), miR-320b (YP02119299), miR-378a-3p (YP00205946), miR-451a (YP02119305), miR-486-5p (YP00204001), and miR-494-3p (YP00204579). The qPCR data were normalized to miR-let7a-5p (YP00205727) values. Relative miRNA expressions were calculated using the 2^−ΔΔCt^ method.

### 4.5. Statistical Analyses

Statistical analyses were performed using GraphPad Prism version 7.03 for Windows (GraphPad Software, La Jolla, CA, USA) and R (version 3.0.2, http://cran.r-project.org/) software, accessed on 22 June 2022. All values are expressed as mean ± SD. After checking whether the data had a normal distribution (Shapiro–Wilk test), Student’s *t*-test, 2-way ANOVA multiple comparison or non-parametric tests (Kruskal–Wallis or Mann–Whitney) were used to compare groups. *** *p* < 0.001, ** *p* < 0.01, and * *p* < 0.05 were considered to indicate significant differences; ns: not significant.

## 5. Conclusions

In summary, our findings demonstrate that LVSELs are LSCs that may contribute to the persistence of CML due to their insensitivity to IM treatment, consequently leading to patient relapse. The differential expression of miR-21 and miR-126 in LVSELs may play a crucial role in their response to IM treatment. It is conceivable that knockdown of miR-21 and miR-126 and/or overexpression of miR-451 might overcome LSC chemotherapy resistance. Furthermore, miRNAs may constitute a new potential biomarker to predict optimal response, thereby allowing the development of specific drugs that target them.

## Figures and Tables

**Figure 1 ijms-24-16671-f001:**
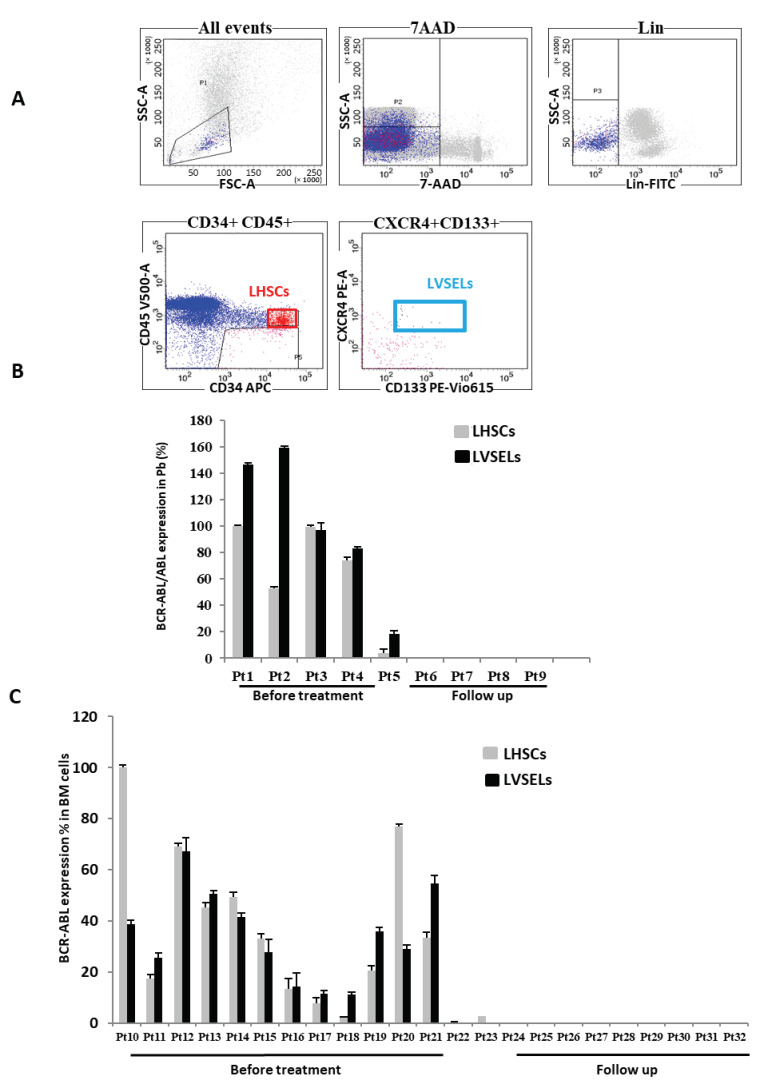
BCR-ABL expression in LHSCs and LVSELs isolated from CML patients and those who are under follow-up. (**A**) Analysis and sorting via FCM of the stem cells Lin^−^CD34^+^CD45^+^ (LHSCs) and Lin^−^CD34^+^CD45^−^ CD133^+^ (LVSELs). (**B**) Real-time RT-PCR analysis of BCR-ABL expression versus ABL expression in peripheral blood and (**C**) in bone marrow.

**Figure 2 ijms-24-16671-f002:**
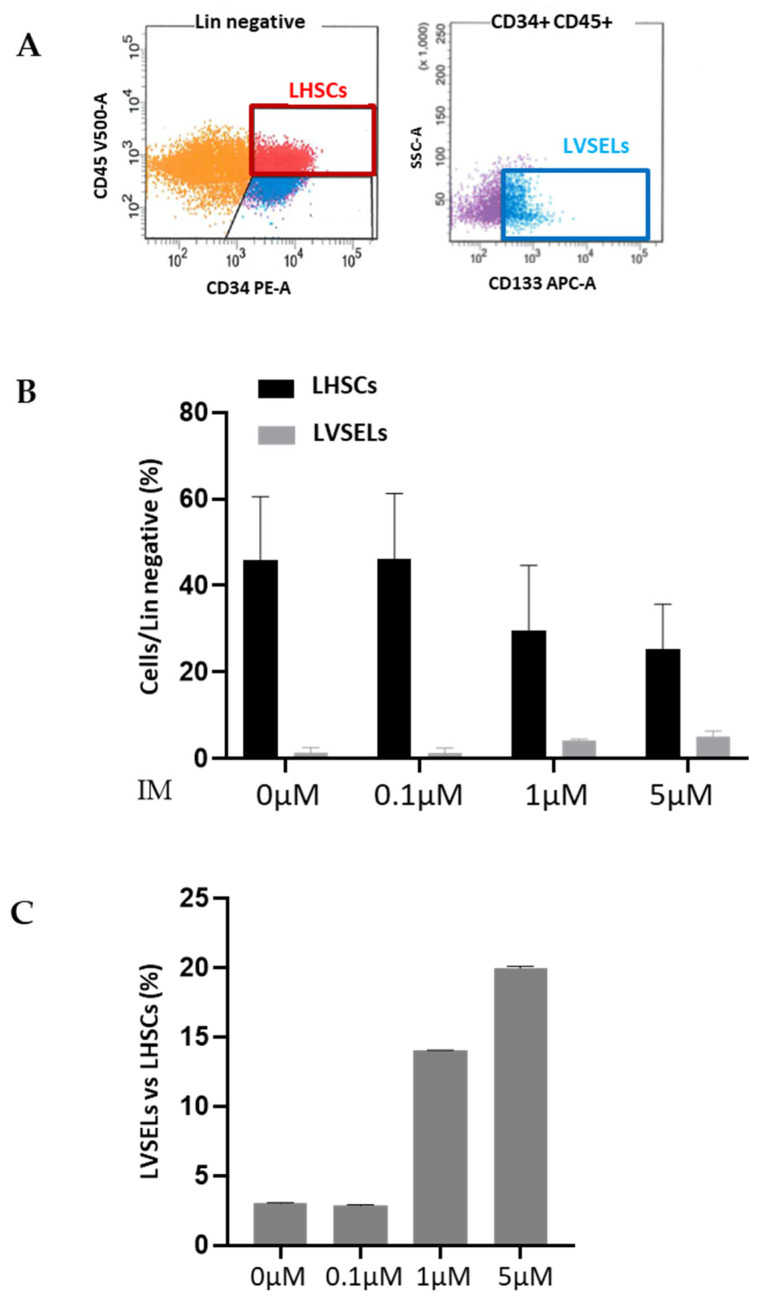
The effects of 48 h of IM treatment on the percentages of LVSELs and LHSCs present in expanded cells in comparison to total lineage-negative cells. (**A**) Dot blot analysis of the 2 subsets of stem cells based on FCM. (**B**) The presence of LVSELs and LHSCs depends on the IM concentration. (**C**) VSEL percentages compared to HSC percentages.

**Figure 3 ijms-24-16671-f003:**
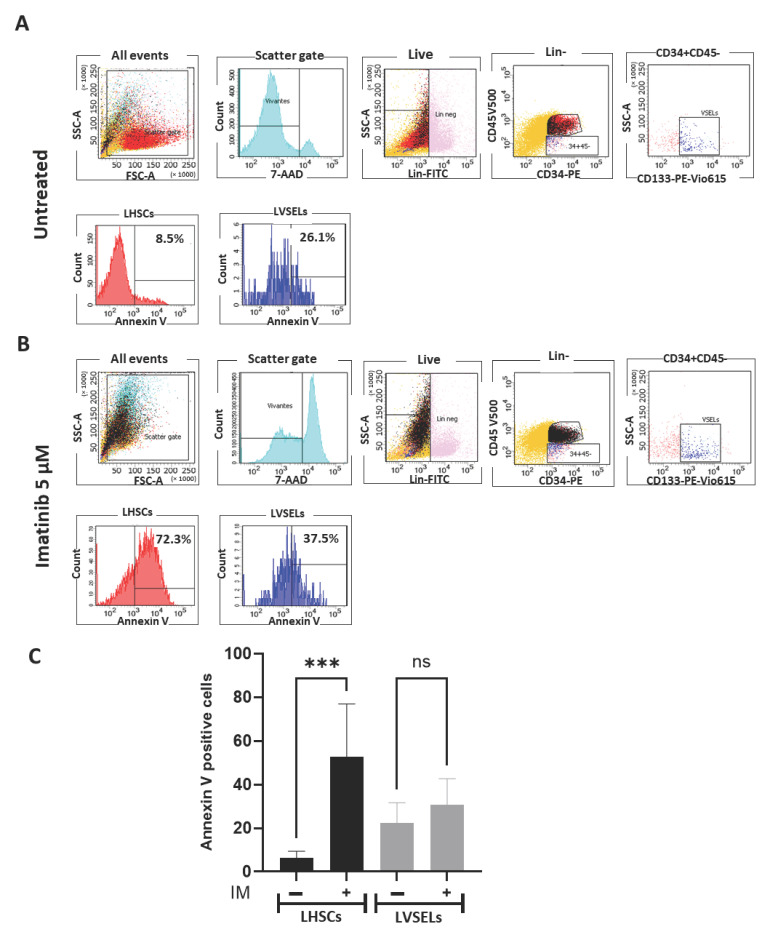
The effects of 48 h of treatment with imatinib on the survival of LHSCs and LVSELs. Evaluation via annexin V labeling of apoptotic LHSCs and LVSELs in IM-untreated (**A**) and IM-treated (**B**) cells (representative experiment). (**C**) The percentages of LHSCs and LVSELs positive for annexin V calculated from 5 independent experiments. Data are presented as means ± SD (*** *p* < 0.001; ns: not significant).

**Figure 4 ijms-24-16671-f004:**
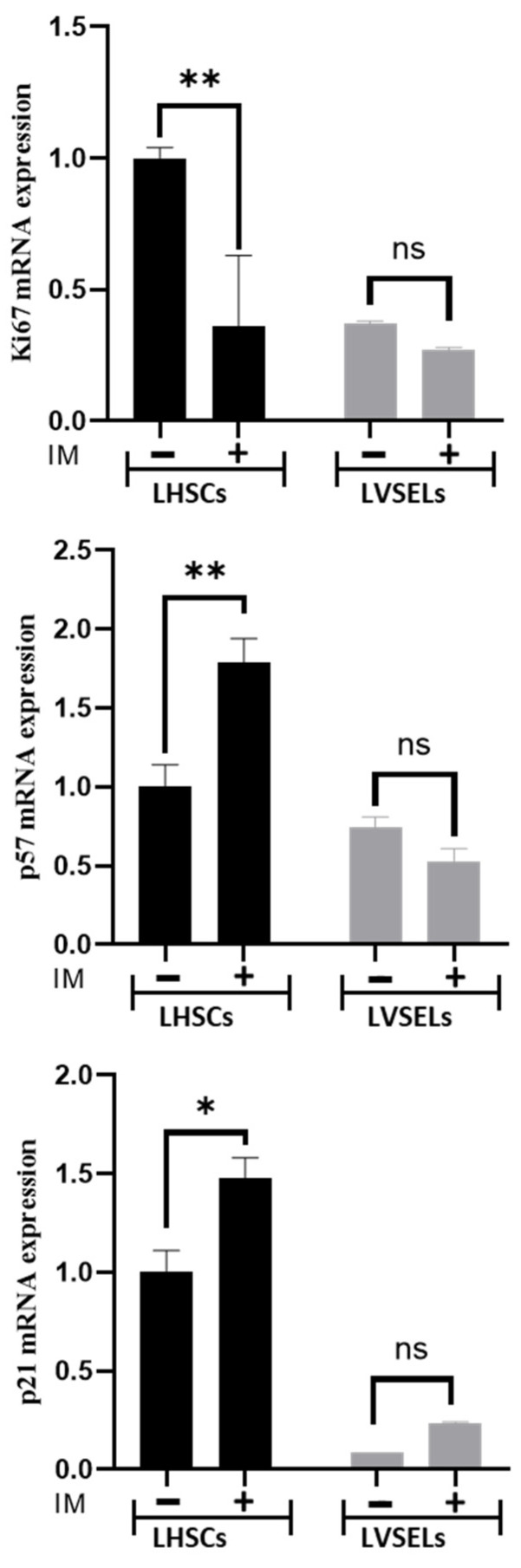
The effects of 48 h of treatment with IM on the indicated gene expression levels in LVSELs and LHSCs as determined using real-time RT-PCR and calculated from 4 independent experiments. Data are presented as means ± SD (** *p* < 0.01, * *p* < 0.05; ns: not significant).

**Figure 5 ijms-24-16671-f005:**
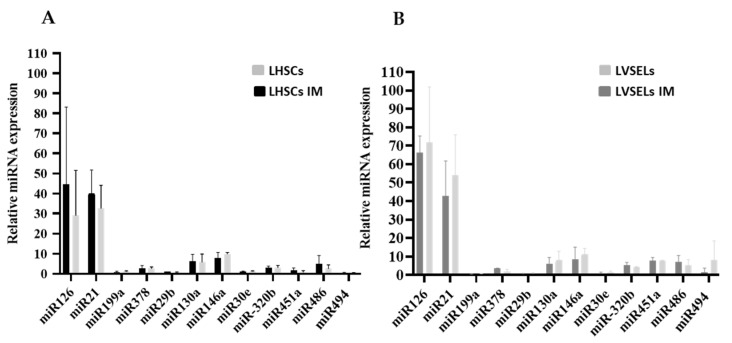
miRNA expression in IM-untreated and IM-treated LHSCs (**A**) and LVSELs (**B**) calculated from 3 independent experiments. Data are presented as means ± SD.

**Figure 6 ijms-24-16671-f006:**
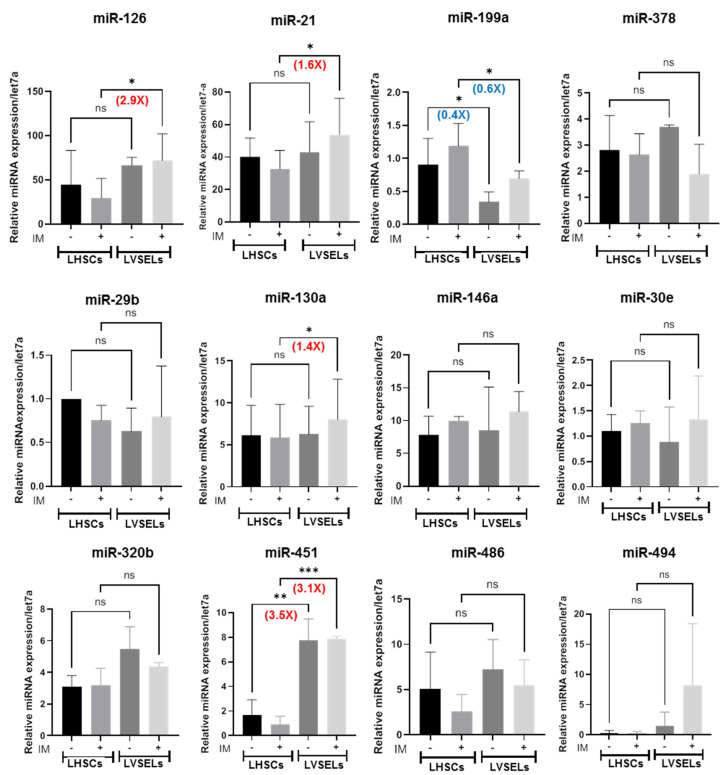
Relative expression of the indicated miRNAs normalized to small RNA (let-7A) values in LHSCs and LVSELs treated (+) or not treated (−) with IM (*n* = 3). The fold changes of significantly upregulated and downregulated miRNA are indicated in red and blue respectively. Data are presented as means ± SD (*** *p* < 0.001, ** *p* < 0.01, * *p* < 0.05; ns not significant).

**Table 1 ijms-24-16671-t001:** Patient characteristics.

	Reference Range (M/F) ^#^	Diagnosis (*n* = 16)	Follow Up (*n* = 20)
Gender (M/F)	-	(10/6)	(14/6)
Age (year)	-	57 ± 16	56 ± 15
WBC (×10^9^/L)	3.9–10.9/4.9–12.68	92 ± 76	6 ± 2
RBC (×10^12^/L)	4.44–5.61/3.92–5.08	4 ± 1	4 ± 1
Hgb (g/dL)	13.5–16.9/11.9–14.6	12 ± 2	13 ± 3
hematocrit	40.0–49.4/36.6–44.0	38 ± 5	38 ± 10
PLT (×10^9^/L)	166–308/173–390	586 ± 560	204 ± 91

M: male; F: female WBC: white blood cells; RBC: Red blood cells, Hgb: hemoglobin, PLT: platelet count. ^#^ The reference range represents the ranges observed in normal male/female subjects.

**Table 2 ijms-24-16671-t002:** Forward, reverse oligonucleotides and probes used for Real time RT-PCR.

Primer Sequences	
BCR ABL M Fw	TCC GCT GAC CAT CAA TAA GGA
ABL Rv	CAC TCA GAC CCT GAG GCT CAA
Probe BCRABL	[6-FAM] CCC TTC AGC GGC CAG TAG CAT CTG A [BHQ1]
ABL1 Fw	TGG AGA TAA CAC TCT AAG CAT AAC TAA AGG
ABL2 Rv	GAT GTA GTT GCT TGG GAC CCA
Probe ABL	[6-FAM] CCA TTT TTG GTT TGG GCT TCA CAC CAT T [BHQ1]
KI67 Fw	ATTGAACCTGCGGAAGAGCTGA
KI67 Rv	GGAGCGCAGGGATATTCCCTTA
P57 Kip2 Fw	GCGGCGATCAAGAAGCTGT
P57 Kip2 Rv	TGGCGAAGAAATCGGAGATCA
p21 Cip1 Fw	TGTCCGTCAGAACCCATGC
p21 Cip1 Rv	AAAGTCGAAGTTCCATCGCTC
GAPDH Fw	CATCGCTCAGACACCATGG
GAPDH Rv	ATGTAGTTGAGGTCAATGAAGGG
β2-microglobulin Fw	TGACTTTGTCACAGCCCAAGATA
β2-microglobulin Rv	AATGCGGCATCTTCAAACCT

## Data Availability

The original contributions presented in the study are included in the article; further inquiries can be directed to the corresponding author.

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
