# Peer review of "Stem Cell Responsiveness to Imatinib in Chronic Myeloid Leukemia"

_ijms, 2023, doi:10.3390/ijms242316671_

Round 1

Reviewer 1 Report

Comments and Suggestions for Authors

General

This is an interesting paper that describes imatibinib (IM) resistance in a subpopulation of CD34+ cells known as very small embryonic-like stem cells (VSELs). The authors demonstrate that the VSELS from CML patients are more resistance to apoptosis during IM treatment. One of the potential mechanisms is that microRNAs (miRs)-451, -126, and -21 might be involved in quiescence and growth of the LSCs. Targeting these miRs could lead to novel approaches to treat CML in patients who are resistant to IM. The strengths of the study include the novel concept, use of primary CML CD34+ cells, and the potential translation to the clinic in the future. Several weaknesses include presentation of the data, lack of clinical information, and need for additional data.

Specific

1.     For Figure 1, it would be helpful to know the clinical features of the CML patients including sex, age, WBC, platelet count, spleen size, etc. The authors should include a Table with this information.

2.     For Figure 2, were these cells identified in patients with IM resistance rather just in culture after 48 hours? This would be important to show.

3.     Figure 3 shows wide errors bars and should be repeated. There might be differences that are not seen in apoptosis in VSELS. Figures 3A and B FAC plots are difficult to read. These figures should be revised.

4.     Figure 4 would have been improved by performing an unbiased analysis of potential genes in VSELS, e.g. by single cell RNA-seq.

5.     Similarly for Figure 5, some of the error bars for miRs are large and should be repeated.

6.     Further characterization of the functional significance of the miRs 126, 451, 21 should be included. They should also be validated in CML patient VSELs.

Reviewer 2 Report

Comments and Suggestions for Authors

Rahlil et al., address the question of the source of imatinib resistance of chronic myeloid leukemia in leukemic very small embryonic-like stem cells (VSELs), which express the BCR-ABL fusion gene. They show that a) leukemic VSELs show higher resistance to imatinib compared to leukemic HSCs (hematopoietic stem cells), b) that after the IM treatment, the expression level of Ki67 RNA was inhibited in leukemic HSCs but not in VSELs, c) the RNA expression levels of the p57 and p21 genes were significantly changed (increased!) after imatinib treatment in leukemic HSCs while not significantly affected in imatinib treated leukemic VSELs, d) there is a differential expression of miRNA after imatinib treatment between the two different cell phenotypes.

A. At some point in the introduction there needs to be a clearer distinction between normal and leukemic cells. In a new paragraph there needs to be a precise definition and corresponding abbreviation for the leukemic stem cell types that authors intend to discuss. The entire article then needs to be subsequently following the given abbreviations (for example LVSEL, LHSC) accurately. Or a clear statement of what means what. HSC in the literature has a certain meaning and it is hematopoietic stem cells that are contrasted to LSC.

B. The paragraph on Ki67 /p57 / p21 is counterintuitively written: “However, the mRNA expression levels of the Cip/Kip family of cyclin-dependent kinase inhibitors, p57 and p21 genes, which are known as important tumor suppressor genes, were significantly reduced after the IM treatment in HSCs” Significantly reduced? This is the opposite to the graph! Furthermore, the use of “However” adds even more to the confusion.

- Interestingly, there is a paper reporting that mitogenic signaling is supressed in VSELs, not referring to neoplastic cells though: PMID: 23708325. Not necessary to cite or to comment, since it does not address neoplastic VSELs.

C. Figure panels 1A, 3A, 3B the size of the letters in the axis labels is somewhat small. If it is possible to improve, this can help.

D. The title “Stem Cell responsiveness to Imatinib in Chronic Myeloid Leukemia” is accurate. However, it does not seem to make readers aware of the discovery of the resistance of VSEL.

Comments on the Quality of English Language

None.

Round 2

Reviewer 1 Report

Comments and Suggestions for Authors

The authors have responded to the previous reviewers' comments and have revised the manuscript.